# The Association Between Physical Activity and Quality of Sleep Among Nursing Students in Saudi Arabia

**DOI:** 10.3390/healthcare13161991

**Published:** 2025-08-14

**Authors:** Eman Bajamal, Jori Alotaibi, Danah Balamash, Esraa Alsaeedi, Hanan Ali, Joud Alzahrani, Layan Swat, Ajwan Alamri, Raneem Jundi, Renad Alzahrani, Samar Alharbi

**Affiliations:** 1College of Nursing-Jeddah (CON-J), King Saud bin Abdulaziz University for Health Sciences (KSAU-HS), King Abdulaziz Medical City, National Guard Health Affairs (NGHA), Mail Code 6565, P.O. Box 9515, Jeddah 21423, Saudi Arabia; alotaibi20545@ksau-hs.edu.sa (J.A.); balamash20550@ksau-hs.edu.sa (D.B.); aisaeedi20505@ksau-hs.edu.sa (E.A.); ali20548@ksau-hs.edu.sa (H.A.); alzhrani20543@ksau-hs.edu.sa (J.A.); swat20626@ksau-hs.edu.sa (L.S.); alamri20501@ksau-hs.edu.sa (A.A.); jundi20565@ksau-hs.edu.sa (R.J.); 421220581@ksau-hs.edu.sa (R.A.); alharbi20589@ksau-hs.edu.sa (S.A.); 2King Abdullah International Medical Research Center (KAIMRC), King Saud bin Abdulaziz University for Health Sciences (KSAU-HS), King Abdulaziz Medical City, National Guard Health Affairs (NGHA), Jeddah 21423, Saudi Arabia

**Keywords:** physical activity, quality of sleep, Saudi Arabia, nursing students, lifestyles

## Abstract

Background: Nursing students usually face excessive academic and clinical demands that negatively impact their sleep quality. Physical activity (PA) has been proposed to enhance sleep, yet few investigations have focused on this correlation within the Saudi environment. The purpose of this research was to determine the relationship between PA and sleep quality in Saudi nursing students. Methodology: A cross-sectional study was conducted among a sample of 554 nursing students from different universities in Saudi Arabia. The International Physical Activity Questionnaire–Short Form (IPAQ-SF) and the Pittsburgh Sleep Quality Index (PSQI) were used to collect data. Snowball sampling was used to recruit the participants through social media. Descriptive statistics, Pearson correlation, and inferential tests were employed for the analysis. Results: Most participants were female (85.1%) and aged 18–34 years (95.5%). LPA levels were reported by 59.6% of students, and only 8.2% engaged in VPA. The majority (91%) reported poor sleep duration (<5 h), and 57.4% had poor sleep efficiency. Overall, 86.7% of students experienced poor sleep quality. Gender and GPA were significantly associated with both PA and sleep quality. Female students and those with lower GPAs were more likely to report LPA and poor sleep. Marital status was also associated with higher levels of VPA. No significant associations were found with age, academic year, region, income, or parental education. A statistically significant positive correlation was found between PA and sleep quality (*r* = 0.192, *p* < 0.001), suggesting that increased PA is modestly associated with better sleep. Conclusion: The research shows a high rate of poor sleep and low physical activity in nursing students, indicating an alarming health trend. Although PA was linked significantly to better sleep, the modest strength indicates the necessity for multi-component interventions. Structured PA programs and sleep hygiene education should be incorporated into nursing curricula by universities to foster students’ well-being and academic performance.

## 1. Introduction

The nursing education discipline is inherently demanding, characterized by the balance of extensive academic coursework and stringent clinical training requirements. Such requirements place substantial psychological stress on nursing students, often leading to disturbances within their daily routines and reduced sleep quality. Long hours of studying, early morning clinical shifts, and the emotional strain involved with patient care have been identified as primary causative factors in sleep pattern interferences. Poor sleep among nursing students has been linked to declining overall health, cognitive function, and the capacity for effective stress management [1]. Irregular schedules and emotionally demanding clinical duties are identified in previous research as significant contributors to poor sleep in this population. Especially, night shift frequency and the number of shifts per week seem to considerably disrupt sleep with long-term consequences, including susceptibility to mental illnesses like depression and anxiety [1,2].

Inadequate rest not only impair academic performance and overall efficiency but may also lead to clinical errors adverse health effects [3]. Interventions promoting sleep hygiene and stress management could mitigate these outcomes and help students maintain performance [4]. Sleep quality is defined as “an individual’s self-satisfaction with all aspects of the sleep experience” ([5], p.144).

Physical activity (PA) is defined as any movement of the body that requires the use of energy and is generated by the skeletal muscles [6], is widely recognized for its role in enhancing well-being and reducing stress. Notably, PA has been found to enhance sleep by facilitating faster sleep onset and promoting more restorative sleep. Exercise may act as a stress-reduction tool that enhances sleep quality and mood, thereby establishing a cycle of physiological and psychological well-being [7,8].

A study conducted in Greater Boston area reported that individuals with higher PA levels generally experienced better sleep, with notable gender differences. Active females reported more sound sleep than less active ones, and increased PA was associated with improved sleep quality and duration on days with higher step counts [9]. This highlights how both habitual and daily fluctuations in PA can influence sleep.

Globally, research has shown that PA is associated with sleep outcomes, although the mechanisms remain complex. For instance, studies from Italy and the U.S. have found links between moderate to vigorous PA and improved sleep satisfaction and duration, while low PA levels were associated with poor outcomes [10,11]. Similarly, research from Croatia and Egypt highlighted associations between PA and multiple sleep domains such as latency, duration, and efficiency, though not always linear [12,13].

In the Middle East, recent studies reflect similar trends. Research in Dubai found that healthcare students had low PA levels and poor sleep quality, with significant associations between recreational PA and sleep efficiency [14]. A study in Qatar showed that only 64.9% of students met WHO PA recommendations, with males being more active than females and access to recreational facilities linked to better PA engagement [15].

In Saudi Arabia, physical inactivity affects an estimated 42–80% of the population [16]. Among university students—particularly females—low PA and poor sleep quality are prevalent, influenced by cultural factors, limited facilities, and lack of awareness [16,17]. Academic demands also shift students’ focus away from PA toward sedentary leisure [18]. While some studies have examined these patterns, few have directly addressed the association between PA and sleep quality among Saudi university students. Most notably, Albikawi (2023) found that female students with higher stress were more likely to suffer insomnia, while those who engaged in PA reported better sleep and lower stress [19]. However, these studies are limited by gender imbalance and a narrow institutional scope.

In light of these identified gaps, the present study aims to determine the association between PA and sleep quality among male and female nursing students in Saudi Arabia. The findings may help to inform tailored interventions to promote sleep health and overall well-being in future health professionals in the Kingdom.

## 2. Methodology

### 2.1. Design and Settings

This study had a cross-sectional design, which is described as one of the types of observational studies that record data at a specific point in time to investigate both outcomes and their corresponding factors in a particular population [20]. This is a frequently applied approach in population surveys and is very handy for the recognition of patterns and associations but not causality. The study was carried out among male and female nursing students from various universities and colleges in the Kingdom of Saudi Arabia. For maximum coverage, all the nursing students from the first-year level to the internship levels were invited to take part in the study. The participants were recruited via various social media networks, such as WhatsApp, Facebook, Email, Telegram, Twitter, Instagram, Snapchat, and TikTok.

The participation was purely voluntary, and no incentives were given, thereby guaranteeing that the respondents took part out of personal interest. Inclusion criteria were clearly defined: the study participants should be current nursing students in Saudi Arabia during the study period, aged 18–25 years, and fluent in Arabic or English. The selected age group is one that is actively engaged in undergraduate or internship-level nursing education and early professional formation. Exclusion criteria were students studying or residing outside of Saudi Arabia, students above 25 years old, practicing nurses, and students in non-nursing disciplines, such as medicine or dentistry. Students with mental disabilities were also excluded to maintain focus on academic enrolled nursing students who were able to provide valid self-report data.

### 2.2. Sample Size

All participants who met inclusion criteria were invited to participate in this study. Those who agreed to voluntarily participate were involved in the study. The sample size was estimated by G*power software, version 3.1.9.6, a statistical software commonly used for a priori power analysis. The calculation was based on a two- tailed Pearson correlation test, which was the primary statistical method used to examine the association between PA and sleep quality. To detect a medium effect size (*r* = 0.3), in accordance with Cohen’s guidelines [21], and with an alpha level (α) of 0.05 and statistical power of 95% (1 − β = 0.95), the minimum required sample size was calculated to be *n* = 159. However, to ensure robustness in the event of additional planned subgroup comparisons (e.g., chi-square tests) and to increase the precision of the estimates, the target sample size was expanded to *n* = 484.To account for an anticipated 10% non-response or incomplete data rate, an additional 49 participants were added to the calculated sample, resulting in a final target sample size of *n* = 533 (484 + 49). This approach aimed to ensure sufficient statistical power and minimize the risk of Type II error across all planned analyses. Ultimately, 554 participants completed the survey.

### 2.3. Sampling Technique

The sampling technique in use in this study involved a snowballing technique, a non-probability sampling technique extensively used in quantitative studies, especially when trying to gain access to populations that may be difficult to access using traditional methods. The approach starts with a small number of initial participants—commonly named “seeds”—that fulfill the inclusion criteria of the study. These participants are then asked to refer colleagues or friends who also meet the inclusion criteria and might be willing to take part in the study. As the referral continues, the sample gradually increases until the required number of participants is attained or data saturation is achieved. While snowball sampling is often linked with hard-to-reach or excluded populations, such as those that are geographically scattered or socially excluded, it can equally be applied effectively in studies of larger populations where networking allows the recruitment of participants. In this context, the approach is especially valuable in academic environments where contact between individuals is frequent and trust has a significant influence on participation. Snowball sampling is recognized as a type of convenience sampling and is usually combined with purposive or quota sampling approaches in order to make certain that participants possess particular, relevant attributes [22].

### 2.4. Data Collection Methods, Instruments Used, Measurements

The data collected in this study were collected using valid and reliable tools that have been validated and found to be reliable in several previous studies. The online questionnaire consists of three main parts. The first part includes five questions that gathered information about participants’ sociodemographic characteristics including: age, gender, parent’s educational level, family income and the geographical area in which the participants reside.

The second part utilized the International Physical Activity Questionnaire-Short Form (IPAQ-SF), designed for global surveillance of PA and inactivity among adults aged 18 to 65 years [23]. The IPAQ categorizes PA into walking (3.3 METs), moderate (4.0 METs), and vigorous (8.0 METs) intensity. MET-minutes per week are calculated using the formula: MET level × minutes/day × days/week. Activity levels are classified as inactive, moderate (≥150 min/week), or vigorous (≥3 days of vigorous activity totalling ≥1500 MET-min/week). The IPAQ is available in short and long versions and has been validated in multiple countries through international reliability studies. It has also been culturally adapted and translated into various languages. This aligns with findings from prior validation studies reporting comparable reliability across IPAQ domains [16,24,25].

The third part of the questionnaire consists of the 19-item Pittsburgh Sleep Quality Index (PSQI) questionnaire which was developed by Buysse et al. (1989) and is used to evaluate overall sleep quality over one month. The items generate seven component scores: sleep duration, sleep efficiency, sleep disturbances, sleep latency, use of sleep medication, daytime functioning, and subjective sleep quality. Each item is rated on a 0–3 scale, and the total score is 0–21; higher scores indicate poorer sleep quality [26].

Moreover, this tool has been modified for other groups as well, including the PSQI-A for Post-Traumatic Stress Disorder (PTSD) related sleep disturbance and PSQI-C for children [27]. Also, it has been translated culturally into different languages. The Arabic version, piloted by Suleiman and Yates (2011), had good internal consistency (Cronbach’s alpha = 0.65) [28]. The PSQI has been very reliable with an overall Cronbach’s alpha of 0.83 [25], whereas the Chinese version reported a coefficient of 0.68 [29].

### 2.5. Data Management and Analysis

After collecting all the data, the researcher entered this data into an Excel database for analysis, version 16.43. The Statistical Package for the Social Sciences (SPSS for Mac, Version 21.0) was used to analyze the data. Various types of analysis were used including: descriptive statistics, such as means, standard deviations, frequencies, and percentages to describe all the variables. Chi-square tests of independence were conducted to examine associations between sociodemographic characteristics and categorical outcomes, specifically physical activity levels and sleep quality classifications. Pearson correlation coefficients were utilized in testing the relationships among continuous variables. The standard alpha level of 0.05 was set for statistical significance.

### 2.6. Ethical Considerations

The scientific approval was obtained from the Institutional Review Board (IRB)—King Abdullah International Medical Research Center (KAIMRC, Approval Code: NRJ24/014/11, Approval Date: 31 October 2024). Several measures were implemented to protect the rights of study participants and ensure confidentiality. Participants were assigned code numbers for identification purposes, and personal data were separated from any documents containing identifying information, such as consent forms. The questionnaire was administered through a web-based platform and were not request any personal identifiers, such as names or IDs, ensuring that responses remain anonymous. The consent form were incorporated as the first section of the online questionnaire. Respondents were required to click “accept” to indicate their consent to participate before proceeding to the questions. The consent form were clearly state that participation is entirely voluntary and that respondents have the right to withdraw from the study or stop completing the questionnaire at any time.

## 3. Results

A total of 554 participants completed the online questionnaire. Descriptive statistics were used to summarize the sociodemographic characteristics of the sample. The participants’ ages ranged from 18 to 25 years or older, with the majority (95.0%) between 18–24 years, and only 3.2% (*n* = 18) aged 25 or older. As shown in Table 1, the majority of the participants were Saudi (96.5%), and female (85.1%). The predominant group of participants were single (94.7%) and resided with their families (84.9%). Geographically, participants were mainly from the Western region (57.6%). Academically, students were well distributed across all years, with the highest representation in the third (24.3%) and second years (23.8%). In terms of GPA, 31.2% reported scores between 4.75 and 5.00. Regarding family income, 41.3% had a monthly household income exceeding 10,000 SAR. Parental education levels varied, with 36.2% of fathers and 39.7% of mothers holding a high school degree. A small proportion of participants did not respond to some demographic items. These minor discrepancies do not compromise the validity of the results, as missing data were anticipated and accounted for in the a priori sample size calculation, which included an additional 10% to ensure adequate power.

### 3.1. Patterns and Levels of PA Among Participants

As shown in Figure 1, walking was the most commonly reported form of PA. While 11.7% of participants reported no walking, the largest group (25.5%) walked six days per week. Walking between one and four days was reported fairly evenly (13.1–14.9%), while only 6.2% walked five days. For moderate physical activity (MPA), 39.2% of participants reported no engagement. Participation decreased with increasing frequency, with 21.1% engaging for one day and just 0.9% for seven days per week. Similarly, vigorous physical activity (VPA) had low overall engagement. Over one-third (38.3%) reported no vigorous activity, and only 1.6% engaged in VPA five days per week. The highest VPA participation was for two days (20.4%).

The results of this study reveal a general pattern of modest sedentary behavior among the participants. Although some of the students registered comparatively low hours of sitting (2 to 3 h), the mean of 4.84 h a day reveals a large number of hours spent on sedentary activities. Additionally, the high percentage of the respondents who had no clue about the sitting time (16.1%) shows the likely absence of awareness in relation to daily inactivity, which calls for focused educational intervention. As shown in Table 2, the highest proportion (59.6%, *n* = 336) had low levels of PA. Moderate levels of physical activity were reported by 30.5% (*n* = 172), while only 8.2% (*n* = 46) reported engaging in VPA. A small proportion of participants (1.8%, *n* = 10) did not report their physical activity level.

### 3.2. Sleep Quality Components and Patterns Based on PSQI Scores

The components of PSQI analysis showed in Table 3 that the subjective sleep quality was good for the majority of the participants, as 68% qualified their sleep as very good or fairly good (Score 0 or 1). Nevertheless, a considerable percentage (30.2%) had poor subjective sleep quality (Score 2 or 3), as represented by a mean score of 2.21 ± 0.77, which signifies mild to moderate dissatisfaction. For sleep latency, most of the participants (57.6%) indicated sleep delays in falling asleep (Scores 2 and 3), and it would appear that initiating sleep is a common problem, with a mean score of 1.65 ± 0.83.

For sleep duration, a significant 91.0% of respondents had a score of 3, meaning that they slept less than 5 h nightly. This gave the highest component mean at 2.83 ± 0.64, with inadequate sleep being highlighted as a significant issue. For sleep efficiency, 57.4% had a score of 3, indicating poor sleep efficiency (≤65%). Despite 29.8% having high sleep efficiency (Score 0), the mean of 1.90 ± 1.36 indicates heterogeneity in the quality of sleep among the sample. Although sleep disturbances were common (with 82.6% scoring 1), the variability in responses was limited. This concentration suggests that most participants experienced minor but regular disruptions during sleep, such as brief awakenings or environmental noise, but not to a severe or clinically significant extent. This pattern is consistent with existing research on sleep quality among university students.

Sleep medication use was low, as 77.8% of the participants had no use (Score 0). Medication use at least once or twice weekly was reported by a mere 5.3%. This item yielded the lowest mean score of 0.26 ± 0.55, reflecting minimal pharmacologic treatment of sleep issues. Daytime dysfunction was reported to a moderate extent: 45.2% had a score of 2 or 3, indicating compromised daily functioning because of inadequate sleep. The average score was 1.40 ± 0.91, indicating mild to moderate effect on daytime activities.

### 3.3. Association Between Sociodemographic Characteristics and PA Level

Chi-square tests of independence was conducted to examine the association between sociodemographic characteristics and physical activity level among nursing students. The results revealed significant associations with gender, marital status, and cumulative GPA. A significant association was found (χ^2^(2) = 7.29, *p* = 0.026). Female students were more likely to engage in LPA (70.8%) compared to males (55.4%), while males had higher rates of MPA and VPA. Moreover, a significant relationship emerged (χ^2^(2) = 7.24, *p* = 0.027). Married students reported higher rates of VPA (20%) compared to singles (6.9%). In addition, a significant association was found (χ^2^(8) = 16.87, *p* = 0.031). Students with higher GPAs (4.75–5.0) were more likely to engage in VPA (11.4%) compared to lower GPA groups.

In contrast, no statistically significant associations were observed between PA levels and the following variables: age, academic level, family income, geographical region, and parent’s education level. These findings suggest that gender, marital status, and academic performance may influence PA engagement among nursing students, while other socioeconomic factors appear to have limited impact.

### 3.4. Association Between Sociodemographic Characteristics and Quality of Sleep

Chi-square tests of independence were conducted to examine the association between sociodemographic characteristics and sleep quality among nursing students. A Chi-square test revealed a statistically significant association between gender and sleep quality (χ^2^(1, *n* = 551) = 6.458, *p* = 0.011), with poor sleep quality reported by 86.7% of females and 13.3% of males, and only one male classified as having good sleep quality. This indicates that female nursing students were more likely to experience poor sleep quality. A significant association was also observed between cumulative GPA and sleep quality (χ^2^(4, *n* = 551) = 10.998, *p* = 0.027), with students who had lower GPAs (below 3.75) showing a slightly higher proportion of poor sleep quality. Although poor sleep was common across all GPA groups, these results highlight the potential link between academic performances and sleep disturbances.

In contrast, no significant associations were found between sleep quality and the following variables: age, marital status, academic year, geographical region, family income, or parental education levels. Regardless of these characteristics, poor sleep quality was consistently reported by the vast majority of participants, suggesting that sleep issues are widespread and may be more closely tied to behavioral and academic factors than to socioeconomic background.

### 3.5. Association Between PA and Sleep Quality

As shown in Table 4, a statistically significant positive correlation was found between PA and sleep quality (*r* = 0.192, *p* < 0.001). It indicates that higher levels of PA are associated with better sleep quality. However, it is important to note that the strength of this correlation is relatively weak, suggesting only a modest association between the two variables, despite the statistical significance.

## 4. Discussion

The current study adds on the growing body of literature emphasizing the relationship between PA and sleep quality, particularly among university students. The cross-sectional design employed in this study is appropriate for identifying associations between variables; however, it limits the ability to infer causality. Therefore, while the findings highlight important trends, they should be interpreted with caution regarding the directionality of the PA–sleep relationship.

Consistent with previous findings, the results of this study demonstrate the increased levels of PA are modestly associated with improved sleep quality. A notably high proportion of participants (79.1%) were identified as having poor sleep quality based on the Pittsburgh Sleep Quality Index (PSQI), a figure that exceeds similar findings in international studies.

For example, a Brazilian study of university students reported that 75.6% had poor sleep quality. Notably, the study identified that students who took part in free-time PA (FTPA) one to seven times weekly had significantly better sleep results compared to those who were inactive [30]. These findings are consistent with the present results in showing the advantages of regular PA in preventing sleep disturbances. Nevertheless, though the two studies confirm the link between activity and better sleep, the prevalence of poor sleep is still unacceptably high in both groups.

Likewise, Merellano-Navarro et al. (2022) studied sleep quality and PA in Chilean physical education students within the COVID-19 pandemic [31]. They found that higher sleep quality was linked to greater levels of PA and that 79.6% of the sample were classified as poor sleepers [31]—very similar to the present study. Both studies further reported prevalent LPA levels. In the present study, 59.6% of the sample were in the LPA category, and this is in line with global trends reported by Bauman et al. (2012), who noted that physical inactivity is a prevailing public health issue, particularly in modern urban areas [32]. The high prevalence of LPA observed is an indicator of an alarming trend that calls for targeted public health interventions to promote more active lifestyles [32].

Regionally, Mahfouz et al. (2020) identified that 63.9% of students in their Saudi sample had poor sleep quality, with no gender differences [16]. They also identified a high level of physical inactivity (62.7%), with females being less active than males [16]. Their results corroborate the present study, with females being more involved in MPA and males reporting more VPA. The present study, however, examined the frequency of walking in more detail, with only 25.5% of students walking six days a week and 11.7% of students reporting no walking at all. This more detailed perspective provides a deeper understanding of the patterns and types of PA among students.

The more general problem of physical inactivity in Saudi Arabia was also noted by Al-Hazzaa (2007), who found 40.6% of Saudi adults to be inactive [33]. The current study’s results generalize this observations among university students, with 39.2% reported no participation in MPA, and 38.3% reported no participation in VPA. These results represent continued patterns of inadequate activity and support the necessity for culturally appropriate, gender-sensitive interventions.

At the global level, a systematic review conducted by Memon et al. (2021) stated widespread physical inactivity among university students in many studies, even though the relationship between PA and sleep quality was not necessarily statistically significant [33]. However, this review supports the overall trend observed in the present study, which revealed a statistically significant but weak positive correlation between PA, particularly walking frequency, and sleep outcomes. While the association in this study (*r* = 0.192, *p* < 0.001) is modest, it nonetheless aligns with broader patterns suggesting that PA may contribute to improved sleep quality when considered alongside other influencing factors [34].

Additionally, research at Taizhou University, China, in 2021, revealed that (31.5%) had very good subjective sleep quality whereas (30.1%) had poor sleep quality. This study revealed that high-intensity and low-intensity PA were both positively associated with better sleep quality [35]. While the Chinese sample had a lower prevalence of poor sleep compared to the present study, both results support PA as an important modifiable determinant of sleep habits.

In conclusion, the current research identifies a worryingly high level of poor sleep quality and low PA levels among Saudi university students, with a particular gender disparity among females, as the majority of participants (85.1%) were female. While these results are generally in line with both regional and international evidence, the specific breakdown of walking habits, PA intensity, and gender variation offers a more subtle contribution to the literature. The observed association between PA and sleep quality, though modest, highlights the potential value of promoting regular PA as part of broader university health strategies aimed at enhancing student well-being. In addition, an awareness programs for students regarding the significance of PA, its benefits to the different domains of their lives, and the necessity to change their sedentary lifestyle in order to improve their sleep quality are needed.

## 5. Limitations

Although this research provides useful information, a number of limitations have to be considered. Firstly, the cross-sectional nature restricts the potential for identifying causal links between PA and sleep quality since it evaluates information at one time point. Moreover, while the sample was predominantly female (85.1%) and from the Western region of Saudi Arabia, which limits the generalizability of the findings. Although this demographic focus provides valuable insight into a traditionally underrepresented group in PA and sleep studies, future research should aim to include a more balanced gender representation and broader geographic coverage. Moreover, the use of self-reported data may introduce response bias, particularly in reporting PA and sleep behaviors.

In addition, the use of self-report data may create response bias in reporting sleep behaviors and PA. The study also did not account for important confounding factors, such as levels of stress, use of electronic devices before bed, use of caffeine, or any existing medical or psychological disorders, that may have the ability to affect PA as well as quality of sleep. Furthermore, although IPAQ and PSQI are standardized tools, local validity and reliability among Saudi university students were not performed in this study, which may have internal validity implications of the findings. Moreover, physical activity aspects of type, intensity, and duration were not analyzed as distinct domains, which might have been more informative about their differential effect on sleep quality. Finally, the lack of subgroup analyses and longitudinal design restricts interpretation of temporal associations and contribute additional to the limitations of generalizability of the results.

## 6. Recommendations

It is highly necessary to boost PA and facilitate sleep hygiene in order to improve overall well-being and academic performance of nursing students. Participation in more frequent PAs, including walking, cycling and running, can improve physical health and alleviate stress levels. Concurrently, it is critical to maintain good sleeping habits; students are encouraged to have 7 to 9 h of good quality sleep each night by developing a sleep-conducive environment, such as lessening noise, lowering light, and refraining from electronic devices prior to sleeping. These sleep tips also contribute to reducing stress and anxiety. In addition, the program will include evidence-based tools for stress reduction, such as deep breathing exercises, mindfulness meditation, and journaling. These strategies can significantly and systemically support emotional regulation, attention and mental well-being. Together these methods provide a strong foundation for the health and academic success of nursing students.

## 7. Implications for Future Practice

According to the noted association between PA and improved sleep quality, the universities administrators and nursing colleges should accord priority to the development of systematic PA programs to suit students’ requirements and schedules. Physical fitness training can be launched within the nursing curricula that can foster healthier lifestyle habits, potentially leading to improved sleep quality and academic and clinical performance. Moreover, university health and counseling services need to advocate and encourage routine exercise as a non-pharmacologic modality for sleep disturbance among nursing students. Future research should adopt longitudinal or multi-component designs to better understand causal pathways and the long-term effectiveness of integrated PA and sleep interventions among diverse student populations.

## Figures and Tables

**Figure 1 healthcare-13-01991-f001:**
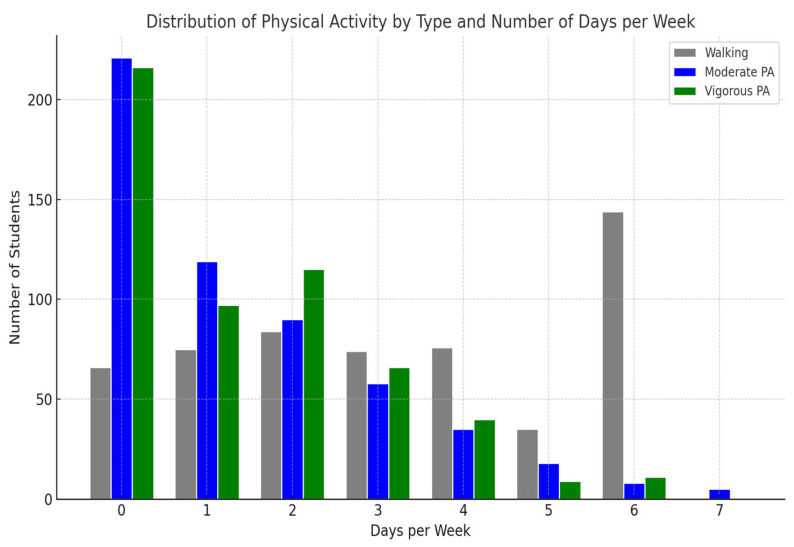
Distribution of Physical Activity by Type and Number of Days per Week (n = 554).

**Table 1 healthcare-13-01991-t001:** Sociodemographic Characteristics of Participants (*n* = 554).

Variable	Category	*n*	*%*
Nationality	Saudi	544	96.5
	Non-Saudi	10	1.80
Age (years)	18–20	273	48.4
	21–24	263	46.6
	≥25	18	3.20
Gender	Female	480	85.1
	Male	74	13.1
Marital Status	Single	534	94.7
	Married	20	3.50
Academic Year	First Year	112	19.9
	Second Year	134	23.8
	Third Year	137	24.3
	Fourth Year	132	23.4
	Intern	39	6.90
Latest GPA	4.75–5.00	176	31.2
	4.50–4.74	119	21.1
	4.25–4.49	91	16.1
	3.75–4.24	122	21.6
	<3.75	46	8.20
Region	Central	96	17.0
	Eastern	38	6.70
	Western	325	57.6
	Southern	65	11.5
	North	30	5.30
Living Arrangement	On-campus	23	4.10
	Off-campus	52	9.20
	With family	479	84.9
Family Monthly Income (SR)	<3000	76	13.5
	3001–5000	83	14.7
	5001–10,000	159	28.2
	>10,000	233	41.3
**Father’s Education**	Uneducated	59	10.5
	High School	204	36.2
	Undergraduate Degree	116	20.6
	Graduate Degree	175	31.0
**Mother’s Education**	Uneducated	78	13.8
	High School	224	39.7
	Undergraduate Degree	93	16.5
	Graduate Degree	159	28.2

Note: GPA = Grade Point Average. SR = Saudi Riyals. Values represent frequencies and valid percentages. Minor deviations from 100% are due to missing responses.

**Table 2 healthcare-13-01991-t002:** Self-Reported Levels of PA among Respondents (*n* = 564).

Category	Frequency (*n*)	Percentage (%)
Low PA	336	59.6
Moderate PA	172	30.5
Vigorous PA	46	8.2
Total	554	98.2
Missing Data	10	1.8
Total	564	100.0

Note. PA = Physical Activity. Values are based on self-reported responses. Percentages may not total 100% exactly due to rounding.

**Table 3 healthcare-13-01991-t003:** Sleep Characteristics of Participants Based on PSQI Components (n = 554).

PSQI Component	Score = 0 (%)	Score = 1 (%)	Score = 2 (%)	Score = 3 (%)	M ± SD
Subjective Sleep Quality	86 (15.2%)	298 (52.8%)	138 (24.5%)	32 (5.70%)	2.21 ± 0.77
Sleep Latency	45 (8.00%)	184 (32.6%)	241 (42.7%)	84 (14.9%)	1.65 ± 0.83
Sleep Duration	21 (3.70%)	13 (2.30%)	7 (1.20%)	513 (91.0%)	2.83 ± 0.64
Sleep Efficiency	168 (29.8%)	40 (7.10%)	19 (3.40%)	324 (57.4%)	1.90 ± 1.36
Sleep Disturbances	68 (12.1%)	466 (82.6%)	15 (2.7%)	5 (0.9%)	0.92 ± 0.42
Use of Sleep Medication	439 (77.8%)	85 (15.1%)	30 (5.3%)	-	0.26 ± 0.55
Daytime Dysfunction	98(17.4%)	201 (35.6%)	186 (33.0%)	69 (12.2%)	1.40 ± 0.91
Global PSQI Score	–	–	–	–	X ± SD

Note. PSQI = Pittsburgh Sleep Quality Index. Each component is scored from 0 (no difficulty) to 3 (severe difficulty). The global PSQI score ranges from 0 to 21, with higher scores indicating poorer sleep quality.

**Table 4 healthcare-13-01991-t004:** Correlation between Physical Activity and Sleep Quality (*n* = 554).

Variables	1	2
1. Physical Activity	—	
2. Sleep Quality	0.192 **	—

Note. ** *p* < 0.001.

## Data Availability

The raw data supporting the conclusions of this article will be made available by the authors upon request.

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
