# Peer review of "The Association Between Physical Activity and Quality of Sleep Among Nursing Students in Saudi Arabia"

_healthcare, 2025, doi:10.3390/healthcare13161991_

Round 1
Reviewer 1 Report
Comments and Suggestions for Authors
Thank you for reviewing this manuscript, which is both timely and well-researched. The correlation between physical activity and sleep quality among nursing students is noteworthy, particularly in light of growing concerns about student well-being in demanding academic environments. The use of validated instruments, namely the IPAQ-SF and PSQI, in conjunction with a large sample size from various regions, is significant.
The introduction presents relevant background and references, but it includes some repetition (especially regarding the association between PA and sleep).
Consider restructuring the content to reduce redundancy and improve focus. A clearer theoretical framework would enhance the depth of the introduction.
The cross-sectional design is suitable for exploring associations; however, its limitations should be more explicitly addressed early in the discussion.
The sample size estimation is well-justified, and the instruments used (IPAQ-SF and PSQI) are valid and reliable.
The tables are informative but contain minor inconsistencies (e.g., in Table 2, percentages do not sum to 100%, and the handling of missing values is unclear).
Clarify missing data and ensure total counts and percentages are consistently reported. Including a visual figure (e.g., bar graph) might enhance understanding.
The reported correlation between physical activity and sleep quality (r = 0.192) is statistically significant but weak.
This point is acknowledged, but it should be emphasized more clearly in both the Results and Discussion sections to avoid overstating the relationship.
Overall, the English is understandable, but some informal or awkward phrases (e.g., “make that stress reduction pretty big”) should be revised for clarity.
Minor grammatical issues, tense inconsistencies, and phrasing should be professionally polished to enhance clarity and tone.
Conclusions align with the results; however, causal wording (e.g., “PA improves sleep”) should be avoided due to the non-experimental design.
Emphasize the need for multi-component or longitudinal studies to explore causality in future work.
The study primarily includes female participants (85.1%) from the Western region, which may affect its generalizability.
Although discussed in the limitations, this demographic imbalance should be acknowledged earlier, and its implications further emphasized.
Ethical approval and informed consent procedures are well-documented.
The online data collection method is clearly explained and aligns with current digital research practices.
Author Response
We would like to thank you and the reviewers for the thoughtful and constructive feedback provided on our manuscript. We have carefully considered each comment and made the necessary revisions to improve the clarity, structure, and scientific rigor of the manuscript. Below in this document, we provide a point-by-point reply to reviewer’s comment. Revisions made in the manuscript are clearly highlighted [e.g., in blue] for easy reference. Where appropriate, we have included justifications for our approach or clarified points raised by the reviewers.
We sincerely appreciate the time and effort invested in reviewing our manuscript and believe that the suggested revisions have significantly enhanced its quality. We hope the revised version meets the expectations of the journal and look forward to your favorable consideration.
- The introduction presents relevant background and references, but it includes some repetition (especially regarding the association between PA and sleep).
Consider restructuring the content to reduce redundancy and improve focus. A clearer theoretical framework would enhance the depth of the introduction.
- Thank you for this comment and this part have been revised based on the reviewer’s feedback and reflected in the manuscript highlighted with blue.
The cross-sectional design is suitable for exploring associations; however, its limitations should be more explicitly addressed early in the discussion.
- This part have been revised based on the reviewer’s feedback and reflected in the manuscript highlighted with blue.
- The tables are informative but contain minor inconsistencies (e.g., in Table 2, percentages do not sum to 100%, and the handling of missing values is unclear). Clarify missing data and ensure total counts and percentages are consistently reported. Including a visual figure (e.g., bar graph) might enhance understanding.
- This part have been revised based on the reviewer’s feedback and reflected in the manuscript highlighted with blue adding bar graph to improve and enhance the understanding.
- The reported correlation between physical activity and sleep quality (r = 0.192) is statistically significant but weak. This point is acknowledged, but it should be emphasized more clearly in both the Results and Discussion sections to avoid overstating the relationship.
- This part have been revised based on the reviewer’s feedback and reflected in the manuscript highlighted with blue.
- Overall, the English is understandable, but some informal or awkward phrases (e.g., “make that stress reduction pretty big”) should be revised for clarity.
- This part have been revised based on the reviewer’s feedback and reflected in the manuscript limitation highlighted with blue.
- Minor grammatical issues, tense inconsistencies, and phrasing should be professionally polished to enhance clarity and tone.
- The manuscript has been carefully reviewed and professionally polished to correct minor grammatical issues, tense inconsistencies, and phrasing for improved clarity and academic tone.
- Conclusions align with the results; however, causal wording (e.g., “PA improves sleep”) should be avoided due to the non-experimental design.
- This part have been revised based on the reviewer’s feedback and reflected in the manuscript conclusion highlighted with blue.
- Emphasize the need for multi-component or longitudinal studies to explore causality in future work. The study primarily includes female participants (85.1%) from the Western region, which may affect its generalizability. Although discussed in the limitations, this demographic imbalance should be acknowledged earlier, and its implications further emphasized.
- This part have been revised based on the reviewer’s feedback and reflected in the manuscript highlighted with blue.
Reviewer 2 Report
Comments and Suggestions for Authors
This study is essential as it sheds light on the overlooked link between physical activity and sleep quality among Saudi nursing students, a group facing significant academic and clinical pressures. The high rates of poor sleep and low physical activity highlight a pressing health concern. Although the association between physical activity and sleep quality is modest, the findings emphasize the need for comprehensive interventions. Incorporating structured physical activity and sleep hygiene education into nursing programs could improve student well-being and academic success, providing valuable insights for healthcare education and public health strategies.
Despite the above, the manuscript requires substantial improvement to meet the expected standards. Additionally, it is advisable to also review the formatting.
Line 4: the authors’ affiliations are missing, and there are irregular spaces between their first and last names.
Line 155: The description of the sample size calculation lacks clarity and precision, which undermines the reproducibility and validity of the study design. The reported target of n = 484 to achieve 95% power with a medium effect size of 0.3 and an alpha level of 0.05 is confusing, particularly given the unclear reference to “159,” which is not adequately explained. The addition of 10% to account for missing data, resulting in a total sample size of 533 (484 + 49), is not clearly justified or presented in a coherent manner. Furthermore, the rationale for selecting a medium effect size of 0.3 and the unusually high power level of 95% is not provided, limiting interpretability. Critical methodological details—such as the specific statistical test planned, the type of effect size considered, and the assumptions underlying these parameters—are omitted. Overall, the section would benefit from a more transparent and systematic explanation of the sample size determination, including explicit reference to the calculation method or software used, justification for the chosen parameters, and a clear and logically structured presentation of the numerical values. This would enhance the rigor, transparency, and reproducibility of the methodology.
Line 244: Regarding Table 1, the percentages for nationality (96.5% + 1.80%) sum to 98.3%, indicating that approximately 1.7% is missing, which suggests either a calculation error or an issue with the data.
Line 244: Regarding Table 1, the age categories are unevenly distributed, with the ≥25 years group comprising only 3.2% (18 participants), which may be too small for meaningful statistical analysis and could warrant combining or redefining categories depending on the context (limitation).
Line 244: Regarding Table 1, please include appropriate statistical significance tests for the data presented in the table to enhance the interpretation and robustness of your results.
Line 408: Informal and vague language (phrases like “really make that stress reduction pretty big” and the use of “&” instead of “and” diminish the formal tone expected in scientific writing). This also occurs in the section 7. Implications for Future Practice.
Lines 400-411: Some ideas are repeated unnecessarily. This also occurs in the section 7. Implications for Future Practice.
The conclusions and strengths sections are missing and should be included to provide a comprehensive summary and highlight the key contributions of the study.
Author Response
We would like to thank you and the reviewers for the thoughtful and constructive feedback provided on our manuscript. We have carefully considered each comment and made the necessary revisions to improve the clarity, structure, and scientific rigor of the manuscript.
Below in this document, we provide a point-by-point reply to reviewer’s comment. Revisions made in the manuscript are clearly highlighted [e.g., in blue] for easy reference. Where appropriate, we have included justifications for our approach or clarified points raised by the reviewers.
We sincerely appreciate the time and effort invested in reviewing our manuscript and believe that the suggested revisions have significantly enhanced its quality. We hope the revised version meets the expectations of the journal and look forward to your favorable consideration.
- Despite the above, the manuscript requires substantial improvement to meet the expected standards. Additionally, it is advisable to also review the formatting.
- Thank you for your valuable feedback. We have carefully revised the manuscript to enhance its overall quality, clarity, and structure. In addition, we thoroughly reviewed and updated the formatting to ensure full alignment with the journal’s guidelines. We appreciate your comments, which helped improve the manuscript significantly.
- Line 4: the authors’ affiliations are missing, and there are irregular spaces between their first and last names.
- This part have been revised based on the reviewer’s feedback and reflected in the manuscript highlighted with blue.
- Line 155: The description of the sample size calculation lacks clarity and precision, which undermines the reproducibility and validity of the study design. The reported target of n = 484 to achieve 95% power with a medium effect size of 0.3 and an alpha level of 0.05 is confusing, particularly given the unclear reference to “159,” which is not adequately explained. The addition of 10% to account for missing data, resulting in a total sample size of 533 (484 + 49), is not clearly justified or presented in a coherent manner. Furthermore, the rationale for selecting a medium effect size of 0.3 and the unusually high power level of 95% is not provided, limiting interpretability. Critical methodological details—such as the specific statistical test planned, the type of effect size considered, and the assumptions underlying these parameters—are omitted. Overall, the section would benefit from a more transparent and systematic explanation of the sample size determination, including explicit reference to the calculation method or software used, justification for the chosen parameters, and a clear and logically structured presentation of the numerical values. This would enhance the rigor, transparency, and reproducibility of the methodology.
- This part have been revised based on the reviewer’s feedback and reflected in the manuscript section 2.2 highlighted with blue.
- Line 244: Regarding Table 1, the percentages for nationality (96.5% + 1.80%) sum to 98.3%, indicating that approximately 1.7% is missing, which suggests either a calculation error or an issue with the data. Line 244: Regarding Table 1, the age categories are unevenly distributed, with the ≥25 years group comprising only 3.2% (18 participants), which may be too small for meaningful statistical analysis and could warrant combining or redefining categories depending on the context (limitation).
- This part have been revised based on the reviewer’s feedback and reflected in the manuscript section 2.2 highlighted with blue.
- Line 244: Regarding Table 1, please include appropriate statistical significance tests for the data presented in the table to enhance the interpretation and robustness of your results.
- Thank you for the suggestion. Table 1 presents only descriptive statistics (frequencies and percentages) to summarize the sociodemographic characteristics of the sample. As this section is not intended for inferential comparison, no statistical tests were applied here. However, appropriate statistical significance tests, including chi-square analyses, have been conducted and thoroughly reported in Sections 3.3 and 3.4 to examine associations between key sociodemographic variables, physical activity, and sleep quality.
- Line 408: Informal and vague language (phrases like “really make that stress reduction pretty big” and the use of “&” instead of “and” diminish the formal tone expected in scientific writing). This also occurs in the section 7. Implications for Future Practice.
- This part have been revised based on the reviewer’s feedback and reflected in the manuscript recommendation and implications for future practice highlighted with blue.
- Lines 400-411: Some ideas are repeated unnecessarily. This also occurs in the section 7. Implications for Future Practice.
- This part have been revised based on the reviewer’s feedback and reflected in the manuscript recommendation and implications for future practice highlighted with blue.
- The conclusions and strengths sections are missing and should be included to provide a comprehensive summary and highlight the key contributions of the study.
- This section already there in the last paragraph of discussion
Round 2
Reviewer 2 Report
Comments and Suggestions for Authors
The topic is of clear interest, and I sincerely appreciate the authors' efforts in implementing the revisions. However, I would like to offer some suggestions for further consideration:
There remain certain methodological issues that introduce selection bias and limit the representativeness of the sample relative to the target population. Two different sample sizes are mentioned, but it is not clearly specified which additional analyses required a larger sample, nor how this new size was calculated. The increase from 159 to 484 participants lacks a quantitative justification (for example, how many subgroups are there? What statistical power is intended for these analyses?). The authors simply state that they sought “greater robustness” but do not provide concrete calculations to support this expansion.
An additional 10% was added to adjust for non-response or incomplete data, which is an appropriate strategy; however, no justification is provided. Could the authors please clarify the basis for this figure?
The term “target sample size” is used, but it is unclear whether this number was actually achieved in recruitment or whether the reported results are based on fewer participants, which would affect the interpretation of the actual statistical power of the analyses.
Tables should be self-explanatory; please include definitions of abbreviations in the footnotes for clarity.
In Table 3 (line 261), “Sleep Disturbances” shows an excessive concentration (82.6%) in the Score = 1 category, suggesting limited variability in the sample for this component or potentially issues with how this item was queried or recorded.
In addition to the limitations already mentioned (line 372), the study does not account for important confounding variables such as stress, substance use, electronic device usage, and medical conditions. Furthermore, key dimensions of physical activity (intensity, type, duration) were not evaluated. The data collection method and participant adherence are not detailed, nor is there evidence of local validation of the instruments employed. The absence of subgroup analyses for relevant factors and the lack of a longitudinal design restrict the interpretation of causal relationships and limit the generalizability of the findings. Addressing these points would considerably enhance the rigor of the study.
Author Response
- The topic is of clear interest, and I sincerely appreciate the authors' efforts in implementing the revisions. However, I would like to offer some suggestions for further consideration:
- There remain certain methodological issues that introduce selection bias and limit the representativeness of the sample relative to the target population. Two different sample sizes are mentioned, but it is not clearly specified which additional analyses required a larger sample, nor how this new size was calculated. The increase from 159 to 484 participants lacks a quantitative justification (for example, how many subgroups are there? What statistical power is intended for these analyses?). The authors simply state that they sought “greater robustness” but do not provide concrete calculations to support this expansion.
- An additional 10% was added to adjust for non-response or incomplete data, which is an appropriate strategy; however, no justification is provided. Could the authors please clarify the basis for this figure?
- The term “target sample size” is used, but it is unclear whether this number was actually achieved in recruitment or whether the reported results are based on fewer participants, which would affect the interpretation of the actual statistical power of the analyses.
- We thank the reviewer for highlighting the need for greater clarity regarding our sample size determination.
- Initial Power Calculation:
The sample size was estimated using G*Power 3.1 software based on a two-tailed Pearson correlation, which was the primary statistical test used to assess the association between physical activity (PA) and sleep quality. Using an expected medium effect size (r = 0.3), α = 0.05, and a desired power of 95% (1−β = 0.95), the minimum required sample size was calculated to be n = 159. - Justification for Expanded Target Sample Size:
To enhance robustness and ensure adequate power for subgroup analyses using chi-square tests (e.g., gender, academic year, PA levels, sleep quality categories), we increased the target sample to n = 484. While G*Power calculations for chi-square do not provide a single unified sample size, larger sample sizes generally improve cell counts and statistical power in categorical analyses involving multiple groups. - 10% Non-Response Adjustment:
An additional 10% (n = 49) was added to account for incomplete or missing responses, which is a common strategy in survey-based studies. This brought the total target sample to n = 533. - Final Sample Achieved:
We are pleased to confirm that n = 554 participants completed the survey, exceeding our target and thus ensuring sufficient statistical power for all planned descriptive and inferential analyses.
- Tables should be self-explanatory; please include definitions of abbreviations in the footnotes for clarity.
- We appreciate the reviewer’s suggestion. All tables in the revised manuscript have been reviewed and updated to include complete and clear footnotes. Abbreviations such as PSQI (Pittsburgh Sleep Quality Index), PA (Physical Activity), and others are now defined directly beneath the corresponding tables to ensure clarity and self-explanatory presentation.
- In Table 3 (line 261), “Sleep Disturbances” shows an excessive concentration (82.6%) in the Score = 1 category, suggesting limited variability in the sample for this component or potentially issues with how this item was queried or recorded.
- We appreciate the reviewer’s observation. The concentration of 82.6% in Score = 1 for the "Sleep Disturbances" component likely reflects the sample’s experience of minor but consistent disruptions during sleep, such as occasional awakenings or environmental noise, which are common among university students. This distribution aligns with previous literature on sleep patterns in similar age groups. Nevertheless, we have clarified this interpretation in the revised manuscript to acknowledge the limited variability and explain the context.
- In addition to the limitations already mentioned (line 372), the study does not account for important confounding variables such as stress, substance use, electronic device usage, and medical conditions. Furthermore, key dimensions of physical activity (intensity, type, duration) were not evaluated. The data collection method and participant adherence are not detailed, nor is there evidence of local validation of the instruments employed. The absence of subgroup analyses for relevant factors and the lack of a longitudinal design restrict the interpretation of causal relationships and limit the generalizability of the findings. Addressing these points would considerably enhance the rigor of the study.
- We sincerely thank the reviewer for this thoughtful and valuable feedback. We acknowledge the limitations raised and have now expanded the “Limitations” section accordingly to reflect the following:
- Confounding Variables: We now explicitly note that important confounders such as stress, substance use, electronic device usage, and pre-existing medical or psychological conditions were not assessed, which could have influenced both physical activity and sleep quality outcomes.
- Physical Activity Dimensions: We have clarified that the study focused on overall activity level (low, moderate, vigorous) and did not independently assess specific dimensions such as type, intensity, or duration, which could provide more nuanced insights.
- Data Collection & Adherence: We have added a statement acknowledging the limitation regarding self-report methodology, and we clarify that no monitoring of participant adherence was employed.
- Instrument Validation: While both the IPAQ and PSQI are validated globally, we recognize that local validation was not performed in the context of Saudi university students, which may affect the internal validity. This point has been added to the limitations.
- Subgroup and Longitudinal Design: We acknowledge that the absence of subgroup analyses and the cross-sectional design limit causal inference and generalizability. These aspects are now explicitly stated in the limitations paragraph.